# Automated Breast Cancer Detection Models Based on Transfer Learning

**DOI:** 10.3390/s22030876

**Published:** 2022-01-24

**Authors:** Madallah Alruwaili, Walaa Gouda

**Affiliations:** College of Computer and Information Sciences, Jouf University, Sakaka 72341, Al Jouf, Saudi Arabia; wgaly@ju.edu.sa

**Keywords:** mammogram, breast cancer, deep learning, ResNet, Nasnet-Mobile, transfer learning, medical imaging

## Abstract

Breast cancer is among the leading causes of mortality for females across the planet. It is essential for the well-being of women to develop early detection and diagnosis techniques. In mammography, focus has contributed to the use of deep learning (DL) models, which have been utilized by radiologists to enhance the needed processes to overcome the shortcomings of human observers. The transfer learning method is being used to distinguish malignant and benign breast cancer by fine-tuning multiple pre-trained models. In this study, we introduce a framework focused on the principle of transfer learning. In addition, a mixture of augmentation strategies were used to prevent overfitting and produce stable outcomes by increasing the number of mammographic images; including several rotation combinations, scaling, and shifting. On the Mammographic Image Analysis Society (MIAS) dataset, the proposed system was evaluated and achieved an accuracy of 89.5% using (residual network-50) ResNet50, and achieved an accuracy of 70% using the Nasnet-Mobile network. The proposed system demonstrated that pre-trained classification networks are significantly more effective and efficient, making them more acceptable for medical imaging, particularly for small training datasets.

## 1. Introduction

Breast cancer is one of the most commonly diagnosed chronic illnesses in women. In 2012, there were approximately 1.17 million new cases globally, which represents one-fourth of all the new cases of cancer in women [1]. It is considered to be the second most diagnosed cancer in women, with the second most cancer deaths all over the world [2]. It can be treated by early discovery, which can significantly decrease breast cancer mortality. Breast cancer screening, mammography, is among the most effective ways to detect cancer at an early stage. Mammography is a low-dose X-ray diagnostic procedure for examining the internals of the breast, and it is currently the recommended approach for screening [3,4]. Despite the fact that mammography has enhanced the screening examinations sensitivities, particularly in dense breasts, the rate of false negatives (FNs) remains significant, owing to the presence of dense tissue that might obscure lesions. Other imaging modalities were considered, including ultrasound, magnetic resonance imaging (MRI), and infrared thermal imaging [5,6].

Masses and calcifications are the most common breast irregularities that may signify breast cancer. In preoperative evaluations of lesion extent and post-treatment surveillance in breast cancer patients, the identification and characterization of masses or calcifications are significant. In the mammogram, masses appear as bright regions of various sizes, margins (micro-lobular, circumscribed, indistinct, spiculated, and obscured), shapes (oval, irregular, round, and lobular), intensities, and contrasts of gray-level that rely on the tissues around them. These masses are called tumors and can be either “malignant”, cancerous, or “benign”, non-cancerous [3]. Calcifications, on the other hand, are small deposits of calcium that appear on mammograms as bright white splotches or spots on the background of the breasts’ soft tissue. Usually, calcifications do not show up on ultrasounds, or on breast MRI images they never show up, while a common finding on mammograms is calcification [7,8].

Several parameters, including poor image quality, the subtle nature of radiologists observations, eye exhaustion, or failure, can lead to missed detection. This leads to the task of automated mass detection and classification for both radiologists and computer-aided diagnosis (CAD) systems difficult. The fundamental problem in this case is the lack of a single method that provides satisfactory results for all images [9]. CAD systems have been built using a variety of machine learning techniques to improve the diagnostics performance of medical imaging for breast cancer. To address the machine learning difficulty, these methods are mainly based on classical classifiers that rely on hand-crafted features. As a result, these procedures are generally complicated, time-consuming, and require the involvement of specialists, particularly in the selection and extraction of characteristics. DL has shown promising results in resolving this problem recently. DL efficiency, however, is dependent on training with large, annotated datasets, which unfortunately, public mammography datasets lack [9,10].

Transfer learning is a DL technique employed in this study that can help with the development of effective appropriate classifiers by transferring information from another area with big databases. A literature review is conducted in this study to identify currently utilized pre-trained models, as well as models that must be field-tested. Then we examined transfer learning (modified ResNet50, (MOD-RES), and Nasnet-Mobile) to see how accurate it was in classifying benign and malignant breast mammography abnormalities. Images from MIAS’s public dataset are used in our proposed system’s training and testing. The Nasnet-Mobile network has not yet been utilized, according to our research findings. MOD-RES, which outperforms state-of-the-art approaches, is also trained as a comparison. 

Throughout this work, authors proposed a crossbred deep learning system for breast cancer classification and prognosis that uses two unique deep learning approaches to accurately detect early breast cancer symptoms from mammographic images. The proposed system has two significant phases: preprocessing, and classification. The preprocessing phase is used to improve the overall contrast of the image in order to make the images more visually appealed. The image is also resized and normalized to suit the size of the training model throughout that process. The classification stage, on the other hand, involves a variety of classifiers, and the most effective classifiers are chosen based on the classification error for each case.

The following are the key contributions of this research:Using various evaluation metrics such as accuracy, precision, recall (sensitivity), specificity, and F1-Score. Extensive comparative comparisons were performed to assess the effectiveness of the proposed systems;Mammograms show radiological indications that are readily detectable symptoms. As a result, deep learning-based methods can be used to automatically analyze mammograms, which significantly reduces analysis time;To fine-tune the weights of pre-trained networks on small datasets, as well as train the weights of networks on large datasets, a customized version of ResNet50 (MOD-RES) and a hybrid version of Nasnet and Mobile net were utilized;To improve the generalization effectiveness of the suggested method and prevent overfitting, a different training protocol assisted by different combinations of training policies (e.g., validation patience, and data augmentation) was used.

The remainder of the paper is laid out as follows: In Section 2, a literature review on the use of transfer learning to the classification of mammography abnormalities is conducted. The suggested experimental system, dataset, and model are provided in Section 3. In Section 4, the results of the experiments are provided. The conclusion of the paper is included in Section 5.

## 2. Related Works

Over the years, there have been numerous attempts to develop an automated methodology for identifying breast cancers from mammographic images. Several authors have employed typical machine learning methodologies, which include preprocessing images, feature extraction, feature selection to reduce the features size, and finally a classification algorithm to achieve the expected result. Transfer learning and Convolutional Neural Network (CNN) models, which are the most effective DL techniques currently in the medical domain, are proven to be superior to traditional methods [11,12]. In current history, DL has indeed been successfully implemented in the field of medicine with impressive outcomes and outstanding performance in different challenges compared with human activity. Various medical imaging systems using transfer learning techniques have also been developed to assist physicians and specialists in effective mammograms diagnosis, care, and follow-up examination [13,14]. 

For example, Z. Hussain et al. [15] introduced how to work around CNNs and transfer learning networks to identify pre-segmented breast abnormalities in mammograms as benign or malignant, using a fusion of transfer learning visual geometry group VGG-16-16 (VGG-16) and data augmentation methods to address the tiny training data obtained from the Digital Mammography Screening Database (DDSM), achieving an accuracy of 88%. Another approach presented by M. Alkhaleefah et al. [10], based on the double-shot transfer learning (DSTL) method, was used to enhance the total performance and accuracy of breast cancer classification pre-trained networks. DSTL uses a large dataset that is similar to the target dataset to fine-tune the learnable parameters (weights and biases) of the pre-trained network. The target dataset is then used to fine-tune the networks.

On the other hand, A. Perre et al. [16] proposed a transfer learning approach using three separate networks (VGG-f, VGG-m and caffe). During the fine-tuning process, the output of these pre-trained CNNs was examined twice; one with image normalization and the other without image normalization to identify abnormalities in mammograms. They tested the output of a support vector machine (SVM) fed with CNN extracted features and the combined the use of feature selection to enhance the CNN feature extraction. Another research presented by A. Khamparial et al. [17] implemented a modified version of VGG (MVGG), residual network, and mobile network. Research results using DDSM dataset demonstrated that the proposed learning model for hybrid transfers (fusion ImageNet and MVGG16) obtained an accuracy of 88.3%, where the epoch numbers equal 15. Just the modified VGG-16 design, on the other hand (MVGG16) provided 80.8% precision, and MobileNet provided 77.2% precision. 

Furthermore, L. Falconi et al. [9] introduced a model including: VGG, ResNet, Xception, and Resnext. Their findings showed that in the CBIS-DDSM dataset, fine-tuning achieved the best classifier efficiency in VGG16 with an AUC value of 0.844. While P. Kaur et al. [18] proposed a new technique, applied to the 322 image Mini-MIAS dataset. A pre-processing framework and integrated feature extraction using K-mean convergence for Speed-Up Robust Features (SURFs) are included. Even during the classification process, a new layer was introduced, which achieved 70% training ratios to 30% Deep Neural Network (DNN) and Multiclass SVM Tests. The outcome demonstrates that the consistency test using K-mean clustering and SVM of the suggested automatic DL approach is better than using a decision tree model. Another approach presented by K. Shaikh [19], which evaluates a CNN model using different datasets (MIAS, DDSM, BancoWeb, and LAPIMO) achieved accuracy equal to 87.5%.

Another implementation presented by Wahab et al. [20] used a pre-trained CNN and applied its learnt parameters to another CNN to classify mitoses. Their proposed method attained precision, recall, and F-measure values of 0.50, 0.80, and 0.621, respectively. Using the film mammography number 3 (BCDR-F03) dataset, Jiang et al. [21] achieved an accuracy of 0.88 using GoogleNet and 0.83 using AlexNet. On the other hand, Cao et al. [22] employed random forest dissimilarity to combine distinct feature groups without any fine-tuning on the source network layers (ResNet125). The dataset “ICIAR 2018” was employed, and classification accuracy was improved to 82.90%. Furthermore, Charan et al. [23] trained a CNN with six convolution layers, four average-pooling layers, three fully connected (FC) layers, and a Softmax (SM) function on a 224 × 224 pixel input picture. The total accuracy of this network was 65% employing MIAS database. 

The work presented by Charan et al. [23] is strikingly similar to ours in that it primarily trained and evaluated the model using images from the MIAS dataset. The proposed system demonstrates remarkable results that are more accurate than existing methods. Furthermore, compared with other models such as ResNet, VGG16, or DenseNet, the proposed improved ResNet50 system is lightweight. In terms of accuracy, our proposed system outperformed existing methods.

## 3. Proposed System

Figure 1 depicts the schematic methodology for the breast cancer detection system, which requires retraining transfer DL approaches (MOD-RES, and Nasnet-Mobile) over pre-processed images in the image datastore to learn discriminative and useful feature representations. At the beginning, the used datastore is described briefly, after that the proposed system’s implementation specifics are discussed; including the proposed pre-processing algorithms, the main design, and the adopted approach’s training methodology. 

### 3.1. MIAS Datasets

In this work, MIAS was used. MIAS is an organization of scientific groups in the UK concerned with mammogram perception. A database of mammographic images collected through London’s Royal Marsden Hospital by J. Suckling [24] has been developed. The archive includes 322 digitized films, and 2.3 GB of 8 mm are available. It also contains a report by radiologists on the tumor types and locations. The database is divided into seven sections, including healthy pairs of images and abnormal instances that included circumscribed masses, micro calcification, ill-defined masses, spiculated lesions, asymmetric densities, and architectural distortion. The overall number of images of malignant and benign mass cases are 51 and 48, respectively, before applying the augmentation methods. 

### 3.2. Image Pre-Processing

This step includes data augmentation, image enhancement, image rescaling, and normalization, among other things. Since the model’s network becomes more sophisticated, the number of parameters to learn increases as well, leading to overfitting. The process when a model learns the training data exceptionally well but fails to generalize effectively to subsequent testing data is known as overfitting. Overfitting is a prevalent issue in DL models, and the risk of coming into it increases when the training dataset is limited, as it was in this work. After that the MIAS dataset was divided into three mutually exclusive sets (e.g., preparing, verification, and evaluating sets) to overcome the overfitting issue created by the small number of training photos. The data augmentation was used to prevent skewed prediction outcomes. Augmented images with corresponding masks such as rotation, reflection, shifting, and scaling were generated for each image in the dataset.

The accuracy of a raw image produced by an electronic detector is simply inadequate, reducing the availability of detection and diagnosis. To improve the quality of mammographic images, image enhancement techniques should be used. Furthermore, training DNNs on top of preprocessed images rather than raw image data will significantly reduce the DNNs’ generalization error and training time. As a result, an appropriate image enhancement technique was proposed to improve the low quality of the images before feeding them into the proposed system. First, the image’s small details, textures, and low contrast were improved using adaptive contrast enhancement based on redistribution of the input image’s lightness values, as shown in Figure 2. Consequently, this approach will improve the visibility of the edges and curves in each part of an image while also enhancing the image’s local contrast. 

Since the images in the dataset are grayscale, the images must replicate three times to obtain an RGB image. We consider it is appropriate to normalize the image to a range of values of 256 gray levels, so that the pixels intensity of all images is normalized between −1 and 1 to ensure that the data are within specific ranges and noise is removed. Normalization has the benefit of ensuring the model is less vulnerable to slight variations in weights and making it easier to optimize.

### 3.3. Proposed Learning Methods

One of the most significant challenges researchers confront when analyzing medical data is the restricted number of available datasets. DL models frequently require a massive amount of data as well as expert data labeling, which are both expensive and time-consuming. The proposed system’s main architecture is based on the transfer learning models. The massive number of structures and hyperparameters to be determined is the most difficult challenge when using DL models (e.g., learning rate, number of batch size, number of frozen layers, and number of epochs, etc.). The effects of various hyperparameters value on the performance of the proposed systems were investigated.

Transfer learning using Nasnet-Mobile, and a modified version of ResNet50 (named MOD-RES) were used to classify breast cancer X-ray images into two categories (Benign and Malignant). In addition, to address the lack of data, we used a transfer learning technique that involved oversampling by duplicating the number of images. Figure 1 shows a schematic representation of the proposed system for the prediction of benign and malignant tumors, including pre-trained Nasnet-Mobile, and MOD-RES models.

#### 3.3.1. Nasnet-Mobile

Since AlexNet gained world attention, the development of the CNN has gone through three phases, the principles which are called Deeper is Better, Architecture Engineering, and AutoML. Nasnet is an extensible CNN model that stands for Neural Search Architecture (NAS) Network. The convolutional neural network Nasnet-Large was trained on over a million photos from the ImageNet collection [25]. The basic concepts differ from typical models such as GoogleNet, and it is expected to lead to a big breakthrough in AI in the near future. It is made up of fundamental building blocks that are tuned using reinforcement learning. A cell is made up of only a few functions and is repeated many times depending on the network’s capacity requirements [26]. Nasnet-Mobile is a mobile version of Nasnet with 12 cells, 5.3 million parameters, and 564 million multiply accumulates [27]. This network has never been utilized before to classify mammographic images, as far as we know.

#### 3.3.2. MOD-RES

In 2015, He K. et al. [28] developed ResNet50, a new a residual learning component to the CNN architecture. A standard layer with a skipped connection compensates the residual unit. The skip connection enables a layer’s input signal to traverse the network by linking it to that layer’s output. As a result of the residual units, an extremely deep 152-layer model was trained, which won the 2015 LSVRC2015 competition. Its innovative residual structure allows for a more straightforward gradient flow and more efficient training. It has a top-five error rate of less than 3.6 percent. ResNet has 34, 50, and 101 layers in other versions.

In this section, we go over the details of a potential solution based on a modified version of the ResNet50 [28] model, as shown in Figure 3. Figure 3a depicts the original ResNet50 model, while in Figure 3b, the latest layers are altered by adding one FC layer as well as replacing both the existing FC layer and Softmax layer to construct the proposed model. The ResNet50 model’s original layers were pre-trained on the ImageNet dataset [25]. As a result, the additional layers will assign random weights firstly, after that the back-propagation technique, which is the basic algorithm for training neural network models, is used to update all model weights throughout training.

In the MOD-RES model, shown in Figure 3b, the first FC was replaced with a new FC layer with size 512 and one FC layer with size 2, number of classes, was added after the replaced FC layer and before the Softmax layer which also was replaced with new the Softmax layer. Based on what was mentioned by Basha, S.S et al. [29] when dealing with small datasets, the network needs more FC layers than when dealing with larger datasets. Any neuron from the previous layer is connected to every other neuron in the next layer in the FC layer, and each value contributes to predicting how well a value fits a given class. The output of the final FC layer is then redirected to an activation function, which calculates the class scores. One of DNN’s most common classifiers is Softmax, which computes the probability distribution of the n output groups through its equations. The only drawback for adding a single FC layer is that it is extremely computationally intensive. 

## 4. Experimental Results

Several sufficiently large experiments were performed on the MIAS dataset to demonstrate the efficiency of the proposed systems and to equate their results to the existing state-of-the-art approaches. The proposed system’s code was written throughout MATLAB R2020b and evaluated on a Windows 10 machine with a Core i7-4650U CPU and 8 GB of RAM. All tests were carried out using an 80% random array of mammographic images as a training collection for the proposed DL systems, according to the proposed training scheme. During the learning process, 10% of the training data was chosen at random and used as a validation set to assess their abilities and save the weight combinations with the highest accuracy value.

The proposed system is pre-trained on the MIAS dataset using the Adam and sigmoid optimizer with a learning rate strategy that decreases the learning rate when learning becomes stagnant for a period (i.e., validation patience). The following hyperparameters were used for training either in the Adam or Sigmoid optimizers: number of epochs = 15, batch size varying from 32 to 128 with a move of double its previous value; patience = 6; and momentum = 0.95. Finally, we incorporate a batch re-balancing strategy to enhance infection form distribution at the batch stage.

The batch normalization was utilized because of its effectiveness in preventing network overfitting. Because several steps in such algorithms contain a degree of randomness, DNN methods always provide outcomes with a degree of variability [30]. As a result, ensemble learning is one technique to increase the performance of DNN algorithms. Throughout this study, we suggest that stacking generalization can be implemented through doing numerous training runs of the same model, which we named the multiple-runs ensemble. 

### 4.1. Assessment Metrices

To evaluate performance, our proposed system is compared with other systems using the following performance metrics:(1)$$\mathrm{Overall}\;\mathrm{Accuracy}=\frac{\mathrm{TP}\;+\;\mathrm{TN}}{\mathrm{TP}\;+\;\mathrm{TN}\;+\;\mathrm{FP}\;+\;\mathrm{FN}}$$
(2)$$\mathrm{Precision}=\frac{\mathrm{TP}}{\mathrm{TP}\;+\;\mathrm{FP}}$$
(3)$$\mathrm{Sensetivity}=\mathrm{Recall}=\frac{\mathrm{TP}}{\mathrm{TP}\;+\;\mathrm{FN}}$$
(4)$$F1-\mathrm{Score}=\frac{2*\;\mathrm{Precision}*\mathrm{Recall}}{\mathrm{Precision}\;+\;\mathrm{Recall}}$$

Included are, true positives (TPs) (sufferers correctly defined as having malignant mass), true negatives (TNs) (sufferers correctly reported as having benign mass), false positives (FPs) (sufferers with benign mass identified as having malignant mass), and FN (sufferers with malignant mass identified as not having the disease).

### 4.2. Results of the Proposed Systems

In this section we report the different experiments results of the proposed systems using the MIAS dataset with 80–20% train-test split. That split is selected, to ensure that execution times were not prohibitive. In the first experiment, we trained the MOD-RES and Nasnet-Mobile models for 15 epochs using 10% of the training set as a validation set, a batch size ranging from 32 up to 128, and a learning rate ranging from 0.0002 up to 0.0008 and freeze the weights of the first 50 layers of the model for MOD-RES model and the first 250 layers for Nasnet-Mobile model. We executed the training three times and monitored the average accuracy measures over the validation set. Table 1, Table 2, Table 3 and Table 4 show the average accuracy of an ensemble of the modified models. As mentioned previously, we built model the ensemble in a way of multiple runs (two runs) to train the same model with the same parameters. An observation that can be made is that the accuracy varies from run to run as the weights are initialized randomly each run, only the best run result is saved and shown in Figure 4, Figure 5, Figure 6 and Figure 7. Comparing the two models, the best achieved accuracy for MOD-RES and Nasnet-Mobile models are 70% for both models.

The obtained results were insufficient; therefore, we devised a new strategy to address the issue: we performed oversampling by duplicating images once before applying augmentation to the images, and then MOD-RES was applied. So that in the second experiment, the MOD-RES model was trained only for 15 epochs using 10% of the training set as a validation set, a batch size of 32, 64, 128, and 265, and a learning rate ranging from 0.0002 up to 0.0008 with freezing the weights of the first 50 layers. The training was executed two times and monitored the average accuracy measures over the validation set. Table 5 and Table 6 show the average accuracy of an ensemble of the MOD-RES model which is 89.5% and 86.8% using the Adam and Sigmoid optimizer, respectively. 

Figure 8 and Figure 9 summarize the previous tables, which display the average accuracy of an ensemble of the MOD-RES model using the Adam optimizer, and Sigmoid optimizer for the best run. It can be noticed that the best result obtained using Adam and Sigmoid optimizers provide the result with accuracy equal to 89.5%, and 86.8% respectively. These results revealed that using oversampling for the images improves the overall accuracy of the proposed model (MOD-RES).

Table 7 shows the best results obtained for using the MOD-RES model with and without oversampling, and the Nasnet-Mobile. The disparity in performance can be explained by the model’s inability to learn the large number of parameters due to the small number of images.

Figure 10 shows that the oversampling + MOD-RES model is used in the efficiency comparison of the proposed system with current state-of-the-art approaches, in addition to its usefulness in leveraging the great strengths of each classifier. These findings bolstered the case for implementing the proposed system in real-world environments to help radiologists diagnose breast infection more accurately using mammograms while also reducing their workload.

### 4.3. Comparison to State-of-the-Art Methods

The proposed system’s performance and reliability are compared with the most recent research in mammogram mass detection systems. In this section, we present the proposed system (oversampling + MOD-RES) outcomes and compare them to existing methods (see Table 8). As revealed in Table 8, the proposed system demonstrates remarkable results that are more accurate than existing methods. Furthermore, compared with other models such as VGG16 or DenseNet, the proposed improved ResNet50 system is lightweight. In terms of accuracy, our proposed system outperformed existing methods. 

## 5. Conclusions

Two reliable and automatic mechanisms for breast cancer diagnosis are presented using mammographic images to differentiate between benign and malignant infected subjects. To improve the intensity of the mammographic image and eliminate any noise, the suggested system employs image enhancement techniques. Two alternative DL approaches, Nasnet-Mobile and MOD-RES were trained on top of preprocessed mammographic images to avoid overfitting and increase the overall capabilities of the proposed DL systems. A mammographic image dataset called the MIAS dataset was utilized to assess the proposed system’s effectiveness. The suggested method outperforms professional radiologists with an overall accuracy of 89.5%, precision of 89.5%, recall of 89.5%, and F1-score of 89.5% using MOD-RES + oversampling, while the overall accuracy reaches 70%, precision of 83.3%, recall of 50%, and an F1-score of 62.5% using Nasnet-Mobile. According to comparative studies, the proposed system (MOD-RES + oversampling) beats existing models.

## Figures and Tables

**Figure 1 sensors-22-00876-f001:**
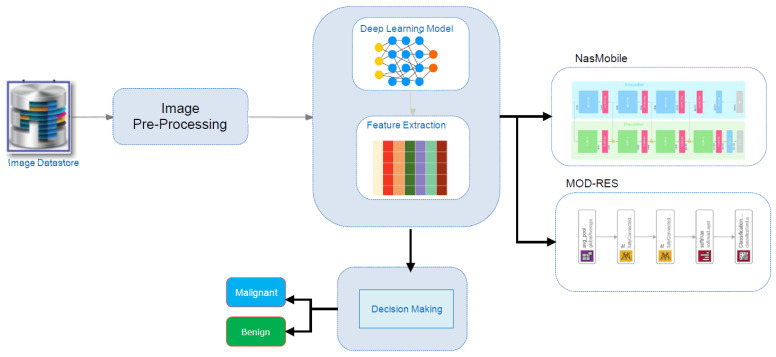
Schematic overview of the proposed system.

**Figure 2 sensors-22-00876-f002:**
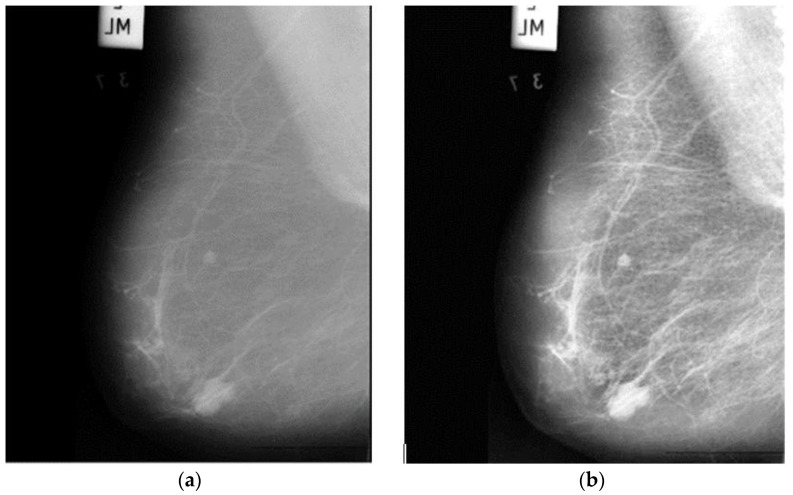
Output of proposed image enhancement process, (**a**) raw image; (**b**) enhanced image.

**Figure 3 sensors-22-00876-f003:**
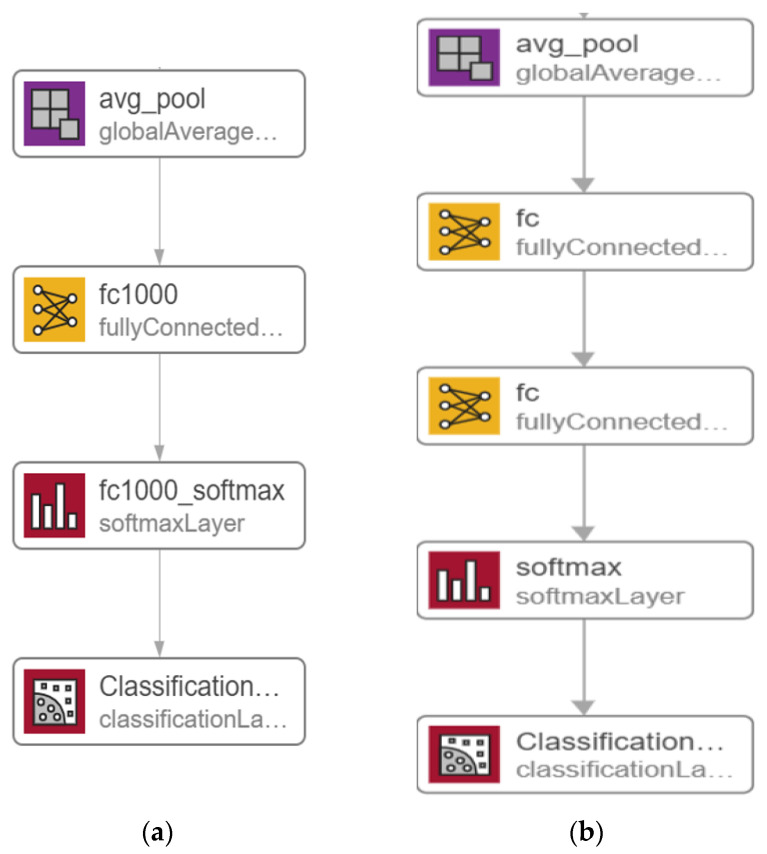
Modified version of proposed ResNet50 model; (**a**) original pre-trained model; (**b**) MOD-RES.

**Figure 4 sensors-22-00876-f004:**
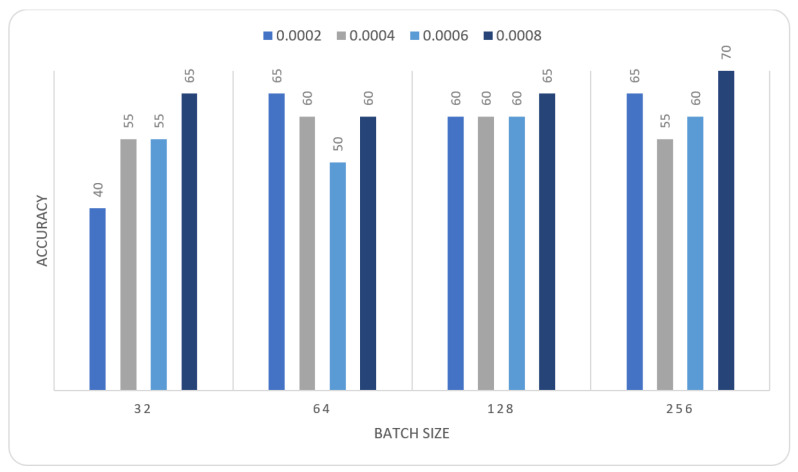
Average accuracy for MOD-RES model with freeze first 50 layers, epochs = 15, optimizer = Adam.

**Figure 5 sensors-22-00876-f005:**
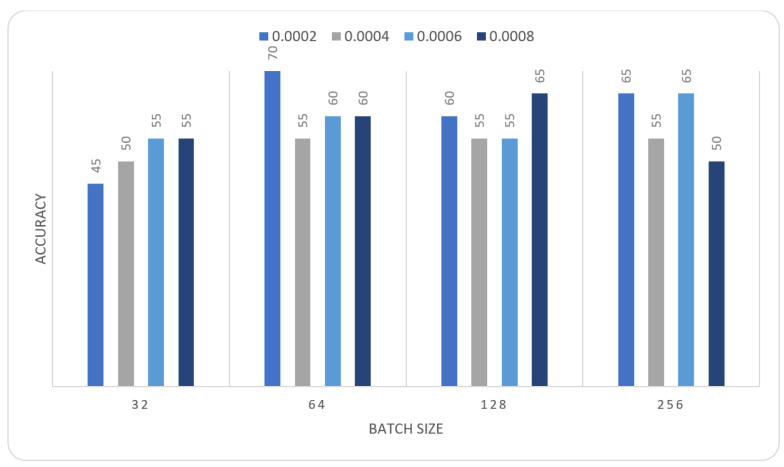
Average accuracy for MOD-RES model with freeze first 50 layers, epochs = 15, optimizer = Sigmoid.

**Figure 6 sensors-22-00876-f006:**
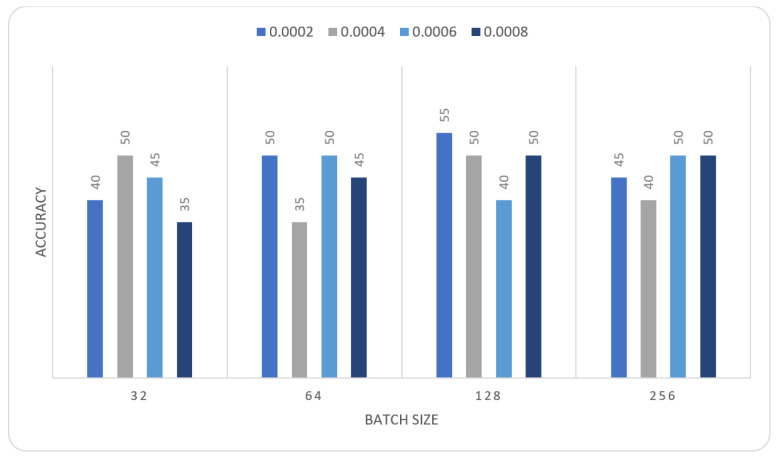
Average accuracy for Nasnet-Mobile model with freeze first 50 layers, epochs = 15, optimizer = Adam.

**Figure 7 sensors-22-00876-f007:**
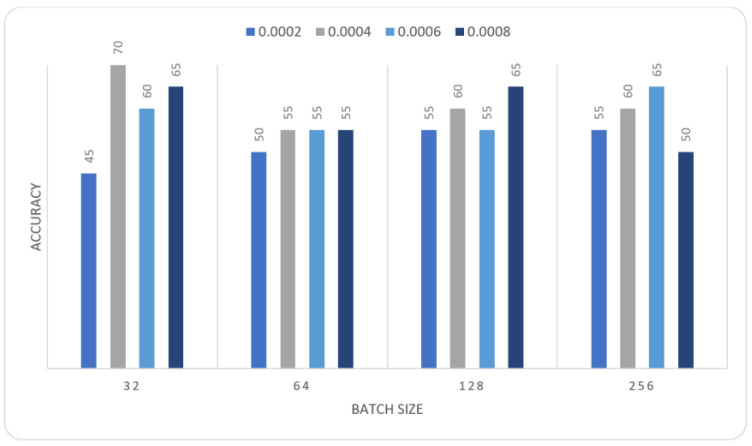
Average accuracy for Nasnet-Mobile model with freeze first 250 layers, epochs = 15, optimizer = Sigmoid.

**Figure 8 sensors-22-00876-f008:**
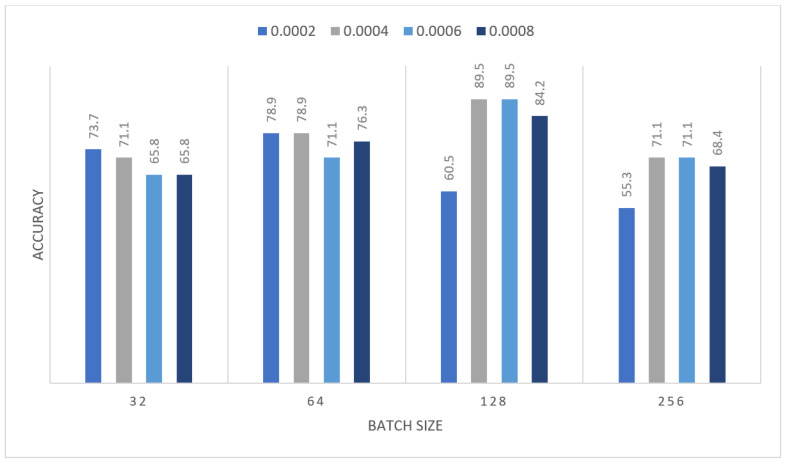
Average accuracy for MOD-RES model with freeze first 50 layers, epochs = 15, optimizer = Adam.

**Figure 9 sensors-22-00876-f009:**
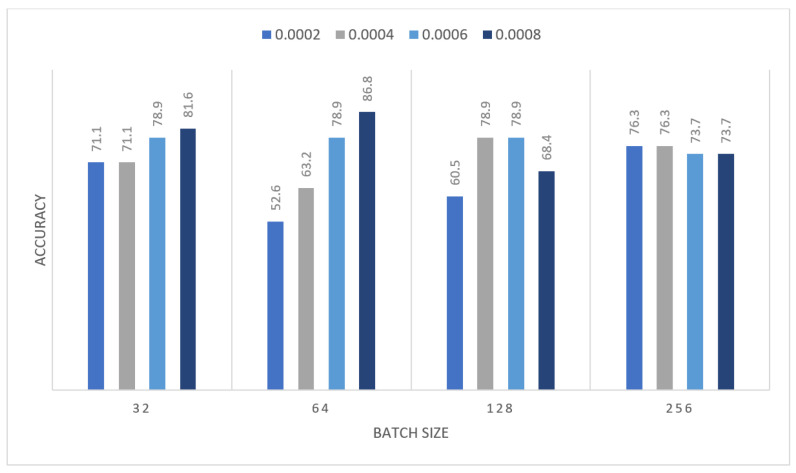
Average accuracy for MOD-RES model with freeze first 50 layers, epochs = 15, optimizer = Sigmoid.

**Figure 10 sensors-22-00876-f010:**
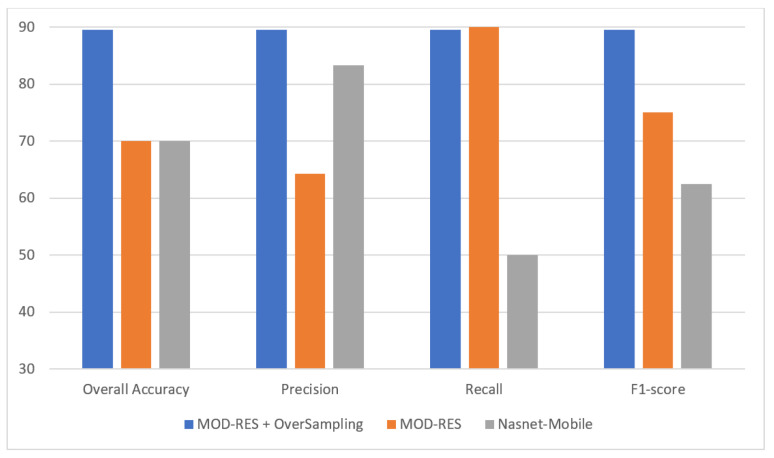
Best result for all models.

**Table 1 sensors-22-00876-t001:** Average accuracy for MOD-RES model with freeze first 50 layers, epochs = 15, optimizer = Adam.

Learning Rate	Ensemble Using Several Runs							
	Batch Size = 32		Batch Size = 64		Batch Size = 128		Batch Size = 265	
	1	2	1	2	1	2	1	2
0.0002	0.4	0.4	0.65	0.55	0.55	0.6	0.45	0.65
0.0004	0.55	0.5	0.6	0.6	0.55	0.6	0.55	0.5
0.0006	0.55	0.45	0.45	0.5	0.6	0.55	0.5	0.6
0.0008	0.55	0.65	0.55	0.6	0.65	0.6	0.65	0.7

**Table 2 sensors-22-00876-t002:** Average accuracy for MOD-RES model with freeze first 50 layers, epochs = 15, optimizer = Sigmoid.

Learning Rate	Ensemble Using Several Runs							
	Batch Size = 32		Batch Size = 64		Batch Size = 128		Batch Size = 265	
	1	2	1	2	1	2	1	2
0.0002	0.45	0.4	0.7	0.5	0.6	0.5	0.55	0.65
0.0004	0.5	0.5	0.55	0.45	0.55	0.55	0.55	0.55
0.0006	0.55	0.5	0.6	0.5	0.55	0.55	0.65	0.5
0.0008	0.45	0.55	0.55	0.6	0.65	0.55	0.5	0.45

**Table 3 sensors-22-00876-t003:** Average accuracy for Nasnet-Mobile model with freeze first 250 layers, epochs = 15, optimizer = Adam.

Learning Rate	Ensemble Using Several Runs							
	Batch Size = 32		Batch Size = 64		Batch Size = 128		Batch Size = 265	
	1	2	1	2	1	2	1	2
0.0002	0.35	0.4	0.4	0.4	0.45	0.5	0.55	0.5
0.0004	0.4	0.45	0.55	0.4	0.6	0.55	0.45	0.45
0.0006	0.35	0.35	0.45	0.5	0.5	0.5	0.5	0.55
0.0008	0.4	0.4	0.55	0.6	0.7	0.6	0.55	0.5

**Table 4 sensors-22-00876-t004:** Average accuracy for Nasnet-Mobile model with freeze first 250 layers, epochs = 15, optimizer = Sigmoid.

Learning Rate	Ensemble Using Several Runs							
	Batch Size = 32		Batch Size = 64		Batch Size = 128		Batch Size = 265	
	1	2	1	2	1	2	1	2
0.0002	0.4	0.4	0.5	0.4	0.55	0.5	0.45	0.4
0.0004	0.35	0.5	0.35	0.35	0.45	0.5	0.4	0.4
0.0006	0.4	0.45	0.5	0.45	0.4	0.4	0.5	0.45
0.0008	0.35	0.35	0.45	0.4	0.5	0.45	0.4	0.5

**Table 5 sensors-22-00876-t005:** Average accuracy for MOD-RES model with freeze first 50 layers, epochs = 15, optimizer = Adam.

Learning Rate	Ensemble Using Several Runs							
	Batch Size = 32		Batch Size = 64		Batch Size = 128		Batch Size = 265	
	1	2	1	2	1	2	1	2
0.0002	0.737	0.711	0.658	0.658	0.553	0.605	0.526	0.711
0.0004	0.5	0.789	0.658	0.711	0.763	0.737	0.632	0.763
0.0006	0.395	0.605	**0.895**	0.842	0.816	0.711	0.658	0.658
0.0008	0.553	0.526	0.711	0.684	0.526	0.789	0.579	0.763

**Table 6 sensors-22-00876-t006:** Average accuracy for MOD-RES model with freeze first 50 layers, epochs = 15, optimizer = Sigmoid.

Learning Rate	Ensemble Using Several Runs							
	Batch Size = 32		Batch Size = 64		Batch Size = 128		Batch Size = 265	
	1	2	1	2	1	2	1	2
0.0002	0.632	0.711	0.684	0.789	0.816	0.737	0.526	0.5
0.0004	0.5	0.526	0.632	0.789	**0.868**	0.737	0.711	0.5
0.0006	0.605	0.526	0.789	0.658	0.684	0.737	0.579	0.474
0.0008	0.658	0.763	0.605	0.737	0.658	0.632	0.579	0.526

**Table 7 sensors-22-00876-t007:** Best result for all models.

Quantitative Measures	MOD-RES Model (Oversampling)	MOD-RES Model	Nasnet-Mobile Model
Overall Accuracy	**89.5**	70	70
Precision	**89.5**	64.3	83.3
Recall	89.5	**90**	50
F1-score	**89.5**	75	62.5

**Table 8 sensors-22-00876-t008:** Comparison of the proposed methodology with state-of-the-art systems.

Recent Work	Technique	Dataset	Number of Images	Accuracy
**Proposed Methodology**	MOD-RES	MIAS	- 51 malignant - 48 benign	89.5%
Charan et al. [23]	CNN	MIAS	- 189 normal - 133 abnormal	65%
Z. Hussain et al. [15]	VGG-16	DDSM	- 1650 mass - 1651 non-mass (normal)	88%
L. Falconi et al. [9]	VGG	CBIS-DDSM	- 912 Benign - 784 Malignant	84.4%
S. Eldin et al. [31]	DenseNet-169	BACH	- 100 normal - 100 benign - 100 in-situ carcinomas - 100 invasive carcinoma	82%
S. Eldin et al. [31]	ResNet50	BACH	- 100 normal - 100 benign - 100 in-situ carcinomas - 100 invasive carcinoma	85%
S. Eldin et al. [31]	ResNet101	BACH	- 100 normal - 100 benign - 100 in-situ carcinomas - 100 invasive carcinoma	88%
S. Siddeeq et al. [32]	ResNet	INbreast	- 115 total images	85.9%
K. Shaikh [19]	CNN	MIAS, DDSM, and BancoWeb LAPIMO	- 211 normal - 110 cancerous	87.5%
S. Salvi and A. Kadam, [33]	CNN	Private Dataset	- 178,059 normal - 70,132 cancerous	87.84%
W. Sun et al. [34]	CNN with Semi-Supervised Learning (SSL) algorithm	full-field digital mammography (FFDM)	3158 region of interests (ROI)	82.43%
Roy et al. [35]	CNN	ICIAR 2018	- 100 normal - 100 benign - 100 in-situ carcinomas - 100 invasive carcinoma	87.4%
S. Alanazi et al. [36]	CNN	Kaggle 162 H and E	- 277,524 total images	87%
S. Singh et al. [37]	Histogram matching (HM) and DL fine-tuning	FFDM	- 830,450 total images	84.7%
K. Mendel et al. [38]	CNN and SVM	FFDM	- 78 total images	89%
A. Rodriguez-Ruiz et al. [39]	CNN	Private Dataset	- 100 total images	88%
M. Yousefi et al. [40]	CCN	Research Laboratory at Massachusetts General Hospital (MGH)	- 5040 total images	87%

## Data Availability

http://peipa.essex.ac.uk/info/mias.html, , accessed on 29 November 2021.

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
