# Peer review of "Automated Breast Cancer Detection Models Based on Transfer Learning"

_sensors, 2022, doi:10.3390/s22030876_

Round 1
Reviewer 1 Report
This study studied the effect of transfer learning strategy on deep learning models, and used a mixture of augmentation strategies to avoid overfitting.
The suggestions are :
- The novelty of this work is relatively low and need to be enhanced.
- The methods and results should be compared and discussed with other state-of-the-art methods in the literature.
Author Response
Reviewer #1
We thank the reviewer for the constructive comment.
Comments and Suggestions for Authors
This study studied the effect of transfer learning strategy on deep learning models, and used a mixture of augmentation strategies to avoid overfitting.
The suggestions are :
- The novelty of this work is relatively low and need to be enhanced.
- The novelty of the presented work lies in:
1- Applying Nasnet-Mobile network for the first time, as according to our known this is the first time for this network to be used
2- enhancing images by applying a different set of filters to enhance the quality of the image before feeding it to the model.
- The methods and results should be compared and discussed with other state-of-the-art methods in the literature.
- Done
Reviewer 2 Report
The authors explored the application of transfer learning application in detecting breast cancer. The domain is investigated by several researchers in the last 5 years. The following comments can further augment the quality of the manuscript:
- Still, the novelty of the work is not clear since the mentioned techniques have been applied and compared with the other researchers already, so how it is different in your case.
- Try to avoid first-person narration like "we conduct..." in scientific writing.
- The related work section is weak.
- Figure 1 is not legible.
- Table 8 shows that with 89.5% accuracy the proposed method is the best, though the authors selective selected the research work, otherwise some other authors claimed more accurate transfer learning techniques for eg. 10.1109/ACCESS.2021.3079204 presented an accuracy of 98.96; so explain this.
Author Response
Reviewer #2
Thanks very much for your comment.
The authors explored the application of transfer learning application in detecting breast cancer. The domain is investigated by several researchers in the last 5 years. The following comments can further augment the quality of the manuscript:
- Still, the novelty of the work is not clear since the mentioned techniques have been applied and compared with the other researchers already, so how it is different in your case.
- The novelty of the presented work lies in:
1- Applying Nasnet-Mobile network for the first time, as according to our known this is the first time for this network to be used
2- enhancing images by applying a different set of filters to enhance the quality of the image before feeding it to the model.
3- using small number of images as well as freezing different number of layers, which certainly affect the performance
4- modifying Resnet-50 architecture by adding Fully connected layer in the later layers to enhance its performance.
5- using Nasnet-Mobile hybrid version for the first time on breast cancer detection
2. Try to avoid first-person narration like "we conduct..." in scientific writing.
- Done
3. The related work section is weak.
- It has been enriched
4. Figure 1 is not legible.
- Done
5. Table 8 shows that with 89.5% accuracy the proposed method is the best, though the authors selective selected the research work, otherwise some other authors claimed more accurate transfer learning techniques for eg. 10.1109/ACCESS.2021.3079204 presented an accuracy of 98.96; so explain this.
- The mentioned paper used all of the 322 images of the dataset with six different abnormality class (calcification (CALC), well-defined circumscribed masses (CIRC), spiculated masses (SPIC), other ill-defined masses (MISC), architectural distortion (ARCH), and asymmetry,) we just use images with mass abnormality (well-defined circumscribed masses (CIRC), spiculated masses (SPIC), other ill-defined masses (MISC). and as shown in table 8 the presented work demonstrates remarkable results that are more accurate than some of existing method.
Reviewer 3 Report
The authors evaluate models based on transfer learning to automatically detect breast cancer. Despite the design under the study is interesting, some issues need to be addressed to make the paper suitable for publication.
First of all, an extensive language editing is required. Some phrases have not sense and/or are hard to read: for example, sentence in rows number 13-15, 153-155, etc… Some other phrases with similar problems are present in the text. Other minor English style issues need to be solved.
Then, what kind of images have been taken in input by networks? Entire images or Region of Interests? The pre-trained networks require images with a fixed size. What is the size in this case? How do the authors resize the images?
What is the difference between Google Net and NASNet? It is not clear from the text.
How many times have the images been duplicated?
Why did not the authors also use the Area Under the Curve (AUC) as an evaluation metric?
In Table 8, the number of images used for each comparative study should be reported.
Author Response
Reviewer #3
Thanks very much for your comment.
The authors evaluate models based on transfer learning to automatically detect breast cancer. Despite the design under the study is interesting, some issues need to be addressed to make the paper suitable for publication.
First of all, an extensive language editing is required. Some phrases have not sense and/or are hard to read: for example, sentence in rows number 13-15, 153-155, etc… Some other phrases with similar problems are present in the text. Other minor English style issues need to be solved.
The paper has been proofread with the help with a native English speaker
Then, what kind of images have been taken in input by networks? Entire images or Region of Interests? The pre-trained networks require images with a fixed size. What is the size in this case? How do the authors resize the images?
The original images were grey scale images with size of 1024*1024, while Resnet50 and Nasnet take RGB image with size 224*244. So in the pre-processing step the size of the image is resized 224 x 224 and converted from grey scale images into RGB images.
What is the difference between Google Net and NASNet? It is not clear from the text.
- GoogLeNet, commonly known as Inception, was developed in 2014 and achieves a top-5 error rate of 5.1% using (single model) and 3.6% using (ensemble). Its architecture was based on a 22-layer deep CNN, however the number of parameters was lowered from 60 million (AlexNet) to 4 million.
- Nasnet, on the other hand, is built on an architectural building block on a tiny dataset that is subsequently transferred to a bigger dataset. Its architecture is built on first searching for the best convolutional layer or cell on CIFAR-10, then applying this cell to the ImageNet by stacking multiple copies of this cell together. Finally, the NASNet model achieves cutting-edge outcomes with a smaller model size and lesser complexity (FLOPs).
- How many times have the images been duplicated? It has been duplicated only once, have been mentioned in the paper
- Why did not the authors also use the Area Under the Curve (AUC) as an evaluation metric? AUC can be calculated form precision and recall, which are calculated in this work
- In Table 8, the number of images used for each comparative study should be reported. Done
Round 2
Reviewer 1 Report
Moderate English changes are required.
Reviewer 2 Report
Required changes have ben made.
Reviewer 3 Report
The authors have addressed to all my concerns